# Effect of Dietary Supplementation of Glycerol Monolaurate on Growth Performance, Digestive Enzymes, Serum Immune and Antioxidant Parameters, and Intestinal Morphology in Black Sea Bream

**DOI:** 10.3390/ani14202963

**Published:** 2024-10-14

**Authors:** Sami Ullah, Jinzhi Zhang, Fengqin Feng, Fei Shen, Mo Qiufen, Jing Wang, Tanzil Ur Rahman, Abdul Haleem, Minjie Zhao, Qingjun Shao

**Affiliations:** 1College of Biosystems Engineering and Food Science, Zhejiang University, Hangzhou 310058, China; 2Zhongyuan Institute, Zhengzhou 450001, China; 3College of Animal Sciences, Zhejiang University, Hangzhou 310058, China; 4College of Food and Health, Zhejiang A & F University, 666 Wusu Street, Hangzhou 311300, China; 5Centre for Animal Sciences and Fisheries, University of Swat, Swat 19200, Pakistan; 6School of Chemistry and Chemical Engineering, Jiangsu University, Zhenjiang 212013, China; 7Ocean Academy, Zhejiang University, Zhoushan 316021, China

**Keywords:** *Acanthopagrus schlegelii*, antioxidant parameters, glycerol monolaurate, growth performance, intestinal development

## Abstract

**Simple Summary:**

Glycerol monolaurate, known for its strong antimicrobial properties, is a chemical compound formed by the combination of lauric acid and glycerol. This research focused on how glycerol monolaurate affects the growth, digestive enzyme activity, immune system, blood antioxidant level, and intestinal structure of black sea bream. This research could contribute to raising healthier aquatic animals, potentially reducing both the costs and environmental impact associated with aquaculture. These findings suggest that glycerol monolaurate may be the most suitable dietary supplement for fish; however, further research on its effect on the gut microbiota and gene expression is still required.

**Abstract:**

An eight-week feeding trial was conducted to examine the impact of dietary supplementation with glycerol monolaurate (GML) on juvenile black sea bream. A basal diet was formulated containing 24% fish meal, while five additional diets were prepared, each supplemented with varying levels of GML: GML1 (0.01%), GML2 (0.02%), GML3 (0.04%), GML4 (0.08%), and GML5 (0.16%). Triplicate tanks were randomly allocated to each diet, each containing 20 fish with an initial weight of 1.55 ± 0.05 g. By the trial’s end, the GML3 group displayed a notably higher final body weight (FBW), weight gain (WG), specific growth rate (SGR), and protein efficiency ratio (PER) compared to the other groups (*p* < 0.05), but the FCR was significantly higher in the control group. However, no significant differences were observed in the MFI, PPV, CF, HSI, IPF, VSI, or SR among the groups (*p* > 0.05). Regarding the proximate compositions of the dorsal muscle and whole body, no substantial differences were observed across the groups (*p* > 0.05). Additionally, there were no significant variations in digestive enzyme activity (*p* > 0.05), serum immune, or biochemical parameters in the midgut and hindgut among the treatment groups. But in the serum immune response IgM, C3 and C4 were significantly higher in the GML3 group as compared to the other groups (*p* < 0.05). However, the GML3 group exhibited significantly greater fore-intestinal villus height, crypt depth, villus height per crypt depth, and the number of goblet cells per villus compared to the other groups (*p* < 0.05). Overall, GML supplementation, particularly GML3, significantly improved growth indicators like the final body weight and intestinal morphology. While certain parameters remained unaffected, these findings suggest GML’s potential as a beneficial dietary supplement in fish diets.

## 1. Introduction

In the past several years, antibiotic treatments have been widely used to mitigate oxidative stress, combat inflammatory diseases, and bolster disease resistance in animals [1]. However, their extensive usage has significantly impacted sustainable development, human health, and the ecological environment, primarily due to the emergence of drug-resistant strains and antibiotic residues [2]. Recognizing these concerns, the European Union (EU) implemented a ban on the inclusion of antibiotics as additives in animal feed since 2006 [3]. Consequently, there has been an urgent push for further research and development of alternative additives capable of replacing antibiotics. Hence, researchers are actively seeking feed additives capable of not only substituting for antibiotics but also enhancing the growth, immunity, antioxidant capacity, gene expression, and intestinal morphology of aquatic animals. Among the potential alternatives, medium-chain fatty acids (MCFAs) have garnered attention. MCFAs represent a class of energy substances with distinct physiological functions and can serve as viable feed additives, offering an alternative to antibiotics [4]. Their potential as antimicrobial agents have sparked interest, given their natural occurrence in foods like coconut oil and their special metabolic functions [5]. As a result, researchers are exploring MCFAs as a possible replacement for lipids in feed materials [6].

Medium-chain fatty acids (MCFAs) are absorbed more quickly from the gastrointestinal tract and transported directly to the liver through the portal vein, whereas long-chain fatty acids (LCFAs) are primarily absorbed via the lymphatic system and enter the peripheral circulation [7] (Figure 1). Medium-chain fatty acids (MCFAs) are taken up by cells independently of membrane transporters and can be directly transported to the mitochondrial intermembrane space without the need for the carnitine shuttle [8]. In the liver, minimal acetyl-CoA enters the citric acid cycle because intermediates such as oxaloacetate and malate are diverted for glucose production. Elevated NADH levels allosterically inhibit the citric acid cycle, leading to reduced cycle activity. Consequently, the metabolism of medium-chain fatty acids promotes ketone production [9]. The rapid absorption and β-oxidation of medium-chain fatty acids (MCFAs) indicate that these fatty acids play a significant physiological role [10]. Additionally, animal studies indicate that medium-chain fatty acids can easily cross the blood–brain barrier and undergo oxidation in the brain [11]. It has been found that MCFA monoglycerides, particularly glycerol monolaurate (GML), have antipathogenic properties [12]. GML, a typical fatty acid glyceride from the group of medium-chain monoglycerides, is easily digestible, efficiently absorbed, and possesses strong antioxidant properties [4]. GML, a nutritional monoglyceride of lauric acid (C12:0) naturally found in coconut oil, is now widely used as a food preservative and emulsifier approved by the US Food and Drug Administration.

GML passes through the gastrointestinal tract with relative stability and a prolonged residence time [14], allowing direct interaction with the gut microbiota, which significantly influences host health and physiology, particularly in metabolism and immune development [15]. Moreover, our recent findings indicated that dietary supplementation with GML promoted the growth of beneficial gut microbiota and had a positive effect on the metabolic system in mice [16]. Recent studies have demonstrated that GML exhibits immunomodulatory functions [17]. It is now widely accepted that symbiotic gut bacteria have a sustained impact on the host’s immune system and metabolism through interactions involving microbial cell components and gene products. Numerous studies have delved into the nutritional physiology of GML in poultry, examining its potential as a feed supplement to enhance productivity and egg quality [18]. GML, a representative fatty acid glyceride of the medium-chain fatty acid monoglycerides, is easily digestible, well-absorbed, and exhibits potent antioxidant properties [4]. Within the liver, GML is efficiently utilized for energy production through mitochondrial beta-oxidation, providing a rapid energy supply [19]. Consequently, GML holds promise as a feed additive to accelerate growth and promote liver lipid metabolism [20]. 

Recent studies have revealed beneficial effects of GML on terrestrial animals such as broilers [21], and weaned lambs [22], significantly enhancing growth, augmenting antioxidant capacity, and mitigating inflammatory responses. However, there remains a substantial research gap concerning GML’s impact on aquatic animals. Limited studies highlight GML’s ability to significantly enhance the growth of *Danio rerio* [23], *Pelodiscus sinensis* [24], and *Larimichthys croceus* [25]. In addition, GML plays a vital role in promoting fat metabolism and reducing fat accumulation in *Salmo salar* [26]. Black sea bream is mostly found in the western Pacific. Due to its meat quality and high tolerance, black sea bream is very popular in southeastern Asia for aquaculture [27].

The aim of this paper was to investigate the effects of GML on growth performance, antioxidant capacity, disease resistance, and inflammatory response in black sea bream. The objective is to enhance the understanding of GML’s applicability in aquatic animals. Our study provides crucial insights into the potential application of GML in promoting the growth, health, and disease resistance of aquatic animals, especially black sea bream, addressing the pressing need for sustainable and effective alternatives in animal health management.

## 2. Materials and Methods

### 2.1. Ethical Statement

The experimental protocols employed in this investigation adhered to the Guidelines of the Care and Use of Laboratory Animals in China. Approval for the study was obtained from the Committee on the Ethics of Animal Experiments at Zhejiang University (Ethics code: ZJU20190052). Stringent measures were implemented to ensure the careful handling of all fish throughout the duration of the experiment

### 2.2. Formulation of Experimental Diets and Their Composition

According to the nutritional requirements of black sea bream, six iso-nitrogenous (41.50%), iso-energetic (19 kJ g^−1^) diets were prepared. These diets were enriched with increasing levels of GML at 0.01%, 0.02%, 0.04%, 0.08%, and 0.16%, labeled as GML1, GML2, GML3, GML4, and GML5, respectively. The GML materials were acquired from South China University of Technology. Primary protein sources such as fishmeal, soybean protein, and meal essence were utilized, while fish oil, soy-lecithin and corn oil were added as lipid sources. Alpha-starch fulfilled the carbohydrate/energy requirements as per the formulation detailed in Table 1. The details of the fatty acid profile are mentioned in Table 2.

Each diet was made from scratch and processed as follows: the raw materials were first crushed into a fine powder, then according to the recipe, the components were precisely weighed and manually stirred for five minutes. Subsequently, the mixed ingredients were transferred to a food mixer for further blending. Additional mixtures of corn, fish oil, and soy lecithin were added to the food mixer after the other ingredients. Gradually, water was added to the mixture and homogenized. The homogenized mixture was passed through a *Φ* 2.5 mm matrix-equipped extruder (Model HUARUI, Wuxi, China; HKJ-218) to form pellets. These pellets were sieved, transferred to airtight containers, and subsequently stored at −20 °C for 72 h to ensure complete drying.

### 2.3. Animal Husbandry, Experimental Site, and Conditions

The young black seabream was sourced from the Marine Fisheries Research Institute of Zhejiang (China). The experimental trials took place at Xixuan Island within the Joint-Laboratory of Nutrition and Feed for Marine Fish, Zhejiang Marine Fisheries Research Institute. Prior to the growth trial, fish were temporarily stored in an indoor tank (10 m × 4 m × 2 m). Black sea bream juveniles were acquired from a nearby hatchery and subjected to a 14-day acclimatization period with a commercial feed containing 42% crude protein (supplied by Ming Hui Co., Ltd., Jiaxing, China) at the rearing facility. Subsequently, 18 cylindrical fiberglass tanks, each with a capacity of 350 L, were utilized. These containers each accommodated 360 fingerlings, all equal in size with an initial body weight (IBW) of 1.55 ± 0.05 g, maintaining a stocking density of 20 fish per tank.

The dietary treatments were allocated randomly in triplicate tanks. Throughout the eight-week rearing period, the fish were provided food three times daily at 8 am, 12 pm, and 16 pm, ensuring satiation. The water underwent purification processes by pumping from the ocean, passing through a sediment pool for 48 h, and further filtration in a sand pool before distribution to each tank. The water flow rate was maintained at approximately 2 L min^−1^, with a consistent seawater temperature of 27 °C ± 1 °C. Continuous aeration using air stones ensured the dissolved oxygen concentration was maintained at >5.0 mg L^−1^. The pH level ranged between 8.1 and 8.3, while salinity was maintained at 28 ± 2 g L^−1^. A 12-h light-dark cycle was maintained, and tanks were cleaned one hour after the last feeding. Fecal collection began in the 6th week of the growth trial. Routine fecal sampling was carried out at 6:00 a.m. following the methodology of [28] and preserved at −20 °C for further analysis.

### 2.4. Sample Collection

At the conclusion of the 56th day of the feeding trial, all experimentally observed fish underwent a 24 h fasting period and were then sedated using tricaine methane sulfonate (60 mg L^−1^). The body weight and length of each fish were quantified. During sample collection, three samples were collected from each tank and each group consisted of triplicate tanks. So, each group had nine samples. Initially, three fish were sampled from each tank for whole-body composition analysis, and following dissection, the liver, viscera, and intraperitoneal fat were weighed and documented to ascertain the body condition indices. And the remaining fish were utilized for the collection of serum, dorsal muscle, and intestine samples, kept at −80 °C.

### 2.5. Chemical Analysis

Blood samples were drawn from the caudal vein of the body using a 1 mL gauge syringe to extract serum. The serum was centrifuged (3000 rpm) at 4 °C for 15 min, and then the serum was then kept at −80 °C. The fish samples underwent proximate analysis using methods outlined by the Association of Official Analytical Chemists [29]. Gut samples were homogenized in ice-cold physiological saline (0.85% *w*/*v*) and subjected to centrifugation at 6000× *g* for 20 min under temperature-controlled conditions. The supernatants obtained were subsequently analyzed for protease, amylase, and lipase activity using diagnostic reagent kits from Nanjing Jincheng Bioengineering Institute (Nanjing, China). Chemical analyses utilized diagnostic reagent kits obtained from Nanjing Jiancheng Bioengineering Institute (Nanjing, China). Total protein (TP), along with albumin (ALB), lysozyme (ZM), alanine transaminase (ALT), aspartate aminotransferase (AST), and total cholesterol T-CHO, were determined as per [30]. Enzyme-linked immunosorbent assay (ELISA) was employed to analyze the concentrations of immunoglobulin (IgM), complement protein C3 (C3), and complement protein C4 (C4) [31]. Intestinal digestive enzymes (trypsin, amylase, and lipase) were quantified following the assay protocols described by [32].

### 2.6. Histological Examination of the Intestines Using Hematoxylin and Eosin Staining (H&E)

For histomorphometric assessment, the intestinal samples underwent sequential dehydration in increasingly concentrated ethyl alcohol solutions, followed by xylene treatment for clearing and subsequent embedding in paraffin wax. Subsequently, the embedded samples were sectioned into 6 μm slices, mounted on glass slides, and stained with hematoxylin and eosin. Three slides were prepared for each intestinal segment for morphological examination, and analysis was performed using a light microscope. This procedure was conducted at the animal physiology laboratory of Zhejiang University in Hangzhou, China. Image acquisition was carried out using an OLYMPUS (CX21) microscope. Villus height measurement was performed using Image-Pro Plus (IPP6.0) software, assessing twelve well-oriented villi per image, with the exact height measured from the villus tip to the crypt junction.

### 2.7. Formulae

Below are the growth performance and feed utilization equations utilized in this study:Initial average body weight (IBW, g)(1)
Final average body weight (FBW, g).(2)
Weight gain rate (WGR, %) = 100 × (Final body weight − Initial body weight)/Initial body weight.(3)
Specific growth rate (SGR, %/day) = 100 × (Natural logarithm of Final body weight − Natural logarithm of Initial body weight)/Number of days.(4)
Mean feed intake (MFI, g fish^−1^ d^−1^) = Dry feed weight in grams/(Fish weight in grams × Days).(5)
Feed conversion ratio (FCR) = Dry feed weight (g)/Wet weight gain (g).(6)
Protein efficiency ratio (PER) = Wet weight gain (g)/Total protein intake (g).(7)
Protein productive value (PPV, %) = 100 × Protein gain (g)/Total protein intake (g).(8)
Condition factor (CF, g cm^−3^) = 100 × [(Final body weight in g)/(Final body length in cm)^3^].(9)
Hepatosomatic index (HSI, %) = 100 × (Liver weight in g/Body weight in g).(10)
Intraperitoneal fat ratio (IPR %) = 100 × (Intraperitoneal fat weight in g/Body weight in g).(11)
Viscerosomatic index (VSI, %) = 100 × (Viscera weight/Body weight).(12)
Survival rate (SR, %) = 100 × (Final fish number/Initial fish number).(13)

### 2.8. Statistical Analysis

The normality of the data was assessed using the Kolmogorov–Smirnov test, and the homogeneity of the data was confirmed with Levene’s test. Mean values ± standard deviations (SDs) were employed to present the results. Data analysis was conducted using IBM SPSS Statistics version 20.0 (IBM, Chicago, IL, USA). One-way ANOVA was employed for data analysis, followed by Tukey’s post -hoc test. A significance level of *p* < 0.05 was considered for determining significant differences.

## 3. Results

### 3.1. Performance in Growth and Efficiency in Utilizing Feed Resources

Table 3 presents the outcomes related to growth performance and the utilization of feed resources. Notably, the GML3 group exhibited significantly improved results (*p* < 0.05) in terms of FB, WG, SGR, and PER when compared to the other groups. However, the FCR was significantly higher in the control group as compared to the treated groups. Conversely, there were no significant differences (*p* > 0.05) observed among all treatment groups for IBW, MFI, PPV, CF, HSI, IPF, VSI, or SR.

### 3.2. Composition of the Whole Body and the Dorsal Muscle

Table 4 provides the basic analysis of the entire body and the dorsal muscles. It is important to note that there were no noteworthy differences in the fat, protein, or ash content of the whole body (*p* > 0.05). However, when it comes to moisture content, the GML1 group had significantly higher levels compared to the other groups. As for the dorsal muscles, there were no significant differences in moisture, lipid, protein, or ash content (*p* > 0.05).

### 3.3. Digestive Enzyme Activity

Table 5 shows the activities of digestive enzymes of juvenile *A. schlegelii*, which presents non-significant differences in trypsin and amylase (*p* > 0.05) across all dietary treatment groups. However, when it comes to lipase, the *control* group had significantly higher levels compared to the other groups (*p* < 0.05).

### 3.4. Indicators of Immune Response and Antioxidant Activity

Table 6 displays data on serum indicators related to the immune system. There were no notable differences (*p* > 0.05) in TP, ALB, LZM, ALT, AST, or T-CHO levels across all treatment groups. However, when it comes to IgM, C3, and C4, the GML3 group had significantly higher levels compared to the other groups (*p* < 0.05).

### 3.5. Immune Parameters in Intestines

Table 7 presents the intestine (hindgut and midgut) immune parameters. The TP, ALB, LZM and T-CHO did not show any significant (p *>* 0.05) variations in the hindgut or midgut in all treated groups.

### 3.6. Intestinal Mucosal Morphology

Table 8 presents the findings regarding intestinal morphometric parameters. Figure 2 displays light microscope images (×100) of the foregut section of the fish. Notably, GML3 supplementation resulted in significantly higher VH, CD, VH/CD, and GC/VH ratios compared to the other groups (*p* < 0.05). As the level of GML supplementation in the fish diet increased from 0.0% to 0.04%, CD, VH, VH/CD, and the number of goblet cells per villus height all showed an upward trend (see Figure 2). Table 7 provides information on the increasing levels of dietary treatments and the corresponding increase in the average number of goblet cells per villus height.

## 4. Discussion

GML is recognized for its multifaceted properties, acting as a fungicide, virucide, anti-inflammatory agent, and antibacterial compound [33]. In the past, adding antimicrobial medications to chicken feed has had a number of detrimental impacts, including changes to the microbiota in the intestines, residues in meat and eggs, environmental contamination, and the development of antibiotic-resistant microorganisms [34]. With increasing public concern about the health risks associated with excessive antibiotic use in animal feed, there is a growing need for the exploration of natural alternatives.

Our study demonstrated that incorporating GML into feed led to increased weight gain, a higher hepatosomatic index (HSI), improved growth performance, and a higher specific growth rate of juvenile black sea bream. Similar outcomes were observed when caprylic and capric acids were used in pigs at a 0.2% dietary supplementation level as reported by Hong, Hwang [35]. These findings indicate that both free medium-chain fatty acids (MCFA) and MCFA bound to triglyceride (at a 2.5% level) in the piglet diet resulted in greater body weight gain and improved feed efficiency compared to the control group, which was fed soybean oil [36]. Additionally, research using pigs showed that GML has substantial potential as a growth stimulator and as a substitute for antibiotics in animal care [37], as observed in our research study.

While the precise mechanism through which GML influences body weight gain remains uncertain, it is speculated to impact meal intake directly or indirectly by altering plasma hormone levels. Our study showed that adding GML to the diet had a positive effect on black sea bream growth, which is consistent with earlier research showing that chain fatty acids might accelerate the growth of young common sea bream [38], tilapia [39], and crucian carp [40]. The PER of black sea bream showed a non-significant variation in feed intake. Large-scale cytokine production consumes considerable energy, leading to increased hepatocyte stimulation for acute phase protein production, resulting in protein loss detrimental to development and disrupting adenosine triphosphate (ATP) production [41]. In this context, GML’s potential to reduce the population of infectious bacteria like *E. coli* aligns with reports by [42], suggesting its pivotal role in maintaining homeostasis.

However, there was no difference in the amount of dietary lauric acid consumed by rainbow trout fed a meal high in coconut oil (31% of total FA, compared to 19% to 29% in the current research) [43]. The same lack of effects on feed intake was reported in hybrid tilapia (*Oreochromis* sp.) fed crude palm kernel oil (lauric acid 46% of total FA) [44]. This implies that the inhibitory effect on feed intake might vary depending on fish species and the source of dietary medium-chain fatty acids (MCFA). According to the PER results, adding GML to the fish diet greatly enhances the way black sea bream uses protein. These results concur with earlier findings published by [45]. Moreover, studies have highlighted the liver as the primary site for chain fatty acid metabolism post-absorption and digestion [46]. In addition to this, there were no significant differences observed in the IPR among the various groups with GML supplementation, aligning with earlier findings in grass carp [47].

Furthermore, this study found that the dietary treatment did not significantly impact the proximate composition of the whole body and dorsal muscle, which aligns with previous research on grass carp and Arctic char [47]. However, this diet induces the effect without cholecystokinin secretion in piglets’ small intestine [48]. Conversely, red drum and African catfish fed CO or LCT did not show a significant effect on body lipid content [49].

Various enzymes in the digestive tract play a vital role in food digestion, leading to improved weight gain and overall fish health. Assessing digestive enzyme activities helps gauge a fish’s nutrient assimilation capacity of a specific diet [50]. Our study found that dietary treatments involving lipase, amylase, and trypsin showed no significant changes, consistent with previous research. Additionally, the supplementation of MCFA and Cuphea seeds increased piglet villus height [14].

To assess the health and nutritional status of black sea bream fingerlings with GML supplementation, various immune, biochemical, and antioxidant parameters were evaluated in this study. These parameters demonstrated that the addition of 0.04% GML had a beneficial impact on the fish’s physiology and immunity in the treated groups. Comparable findings were noted in earlier studies involving various poultry breeds [51], along with a more robust immune system [52]. The elevated TP in this study suggests improved protein metabolism [53].

Serum biochemical measures such as AST, ALB, ALT, AST, and T-CHO did not alter considerably, indicating that the kidney and liver were not severely affected by GML exposure. Hepatocellular injury is indicated by a change in the blood levels of the enzymes AST and ALT [54]. Similar results were reported previously in rats treated with virgin coconut oil [55].

In teleost fish, IgM serves multiple immune functions, including the activation of complement, leading to the elimination or neutralization of bacterial pathogens through specific antigen responses [56]. In this study, no significant difference was observed in serum total protein (TP) or albumin (ALB) levels across all treated groups. However, there was a significant increase in complement protein 3 (C3), complement protein 4 (C4), and immunoglobulin M (IgM) levels in the group supplemented with GML compared to the control group. These findings align with previous research findings [57]. Moreover, in our study, the serum cholesterol levels in all groups exhibited no noteworthy distinctions, which is consistent with prior findings [58].

The intestinal epithelium has a rapid turnover rate, renewing itself every 4 to 5 days, and is arranged into villi-crypt units to maximize absorptive surface and achieve nutritional absorption [59]. Cells from the crypt migrate upward to create villi, with cell shedding occurring at the villi tip and apoptosis primarily in the crypt depth [60]. Notably, post-weaning piglets with significantly shorter villi exhibited decreased digestive capacity [60]. Similarly, providing MCT to suckling piglets improved the duodenal and jejunal villi length, the villi-crypt ratio, and performance. Yet, research on growing broilers is currently lacking [61].

The structural integrity of the intestinal mucosa barrier is crucial for animals to perform at their best since it plays a crucial role in digestion, nutritional absorption, and immune function [62]. This barrier comprises epithelial cells, which are involved in digestion and absorption, and goblet cells, which create mucus to lubricate food and move various components between the layers while also protecting the underlying layers [63]. In our study, fish receiving dietary GML from 0.01% to 0.04% exhibited notably increased VH, CD, VH/CD, and GC/VH values, displaying a well-organized microvilli structure compared to the control group. These findings suggest that GML additives can enhance intestinal integrity and nutrient absorption capacity [64]. Similar enhancements in intestinal morphology were observed in piglets and broiler chickens supplemented with soybean-based diets, emphasizing increased villus height, crypt depth, and mucosal thickness [65]. Reports also indicate that organic acids in diverse diets contribute to improvements in villus height, reduced crypt depth, and increased goblet cell numbers [66]. In our investigation, fish receiving GML supplements exhibited improved villus height, echoing previous findings in pigs receiving MCT supplementation [61]. Comparable findings were also documented in swine when employing medium-chain fatty acids (MCFAs). They demonstrated a notable increase in small intestine villi length, along with a reduction in crypt depth and intraepithelial lymphocyte count [14]. Past research indicated that both caprylic and capric acids, whether administered together or individually, substantially boosted piglet body weight and enhanced villus height [67].

Our research underscores that supplementing black sea bream diets with GML dramatically improved the immune system. Goblet cells, integral to the intestinal immune cell defense [68], were notably increased in number with GML supplementation, indicating improved intestinal growth and functionality in juvenile black sea bream. These reports indicate a direct relationship between weight increase and the well-being of villi in piglets [69]. Taken together, our findings showcase a strong correlation between enhanced intestinal morphology and improved growth, underscoring the vital role of GML in fostering intestinal health and supporting growth in fish.

## 5. Conclusions

In conclusion, the present study indicates that the proper amount of dietary GML in a high SBM-based diet had the potential to enhance growth performance, immune response and intestinal mucosal morphology. The current findings suggested that GML is very useful and the 0.04% level was the best dose in our study. Further, the inclusion of GML in the diet led to significant improvements in villus height, crypt depth, villus height per crypt depth, and the number of goblet cells per villus height in the fore intestinal section. The intrinsic links between the regulatory pathways and the physiological functions of actual mechanisms of GML require molecular-based studies that are suggested for further clarification.

## Figures and Tables

**Figure 1 animals-14-02963-f001:**
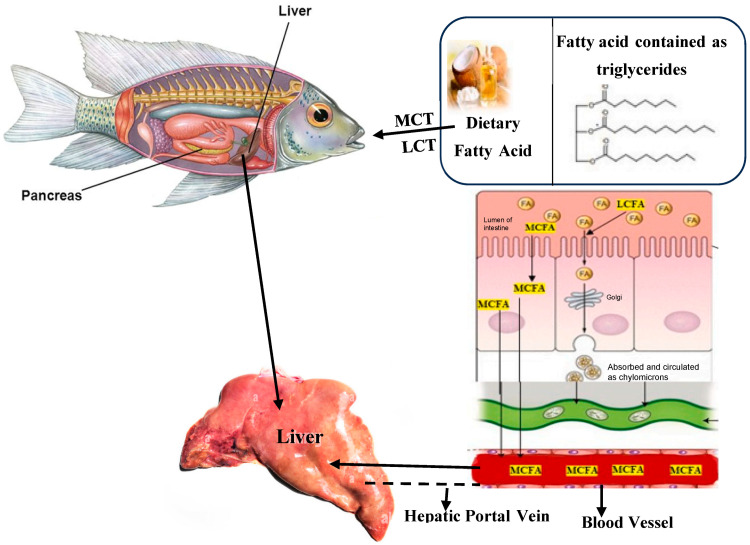
Comparison of the absorption of medium-chain fatty acids (MCFAs) with that of other common long-chain dietary fatty acids. Most common long-chain fatty acids are distributed throughout the body as chylomicrons via the lymphatic and peripheral circulation, whereas medium-chain fatty acids (MCFAs) are primarily absorbed directly into the liver through the hepatic portal vein. MCT refers to medium-chain triglycerides, LCT to long-chain triglycerides, FA to fatty acids, and LCFA to long-chain fatty acids (Adopted from Ref. [13] with permission).

**Figure 2 animals-14-02963-f002:**
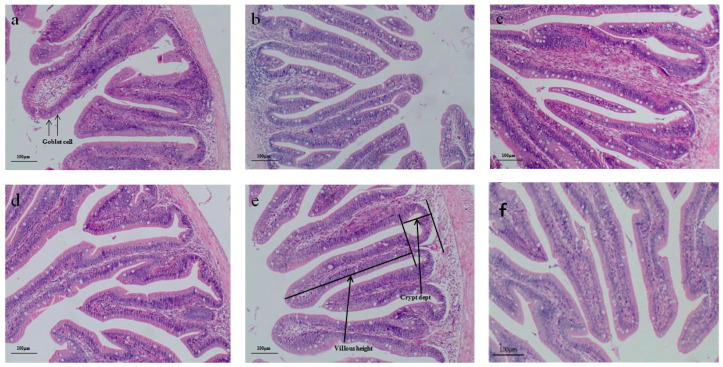
Histology (H&E) of foregut villus structure of *A. schlegelii* fed the experimental diets (×100). Notes: (**a**) Fish fed control diet, 0.0%; (b) fish fed diet GML1, 0.01%; (**c**) fish fed diet GML2, 0.02%; (**d**) fish fed diet GML3, 0.04%; (**e**) fish fed diet GML4, 0.08%; (**f**) fish fed diet GML5, 0.16%. Image (**a**) indicating condensed villus height (VH), fewer goblet cells (GC), and smaller crypt depth (CD); images (**b**–**f**) have longer villi, more goblet cells, and greater crypt depth.

**Table 1 animals-14-02963-t001:** Experimental diets containing different levels of glycerol monolaurate (GML): formulation and their proximate composition (%).

Ingredient	Diets					
	Control	GML1	GML2	GML3	GML4	GML5
FM	19.9	19.9	19.9	19.9	19.9	19.9
SBM	43.5	43.5	43.5	43.5	43.5	43.5
Soy protein concentration	7	7	7	7	7	7
Squid liver meal	3	3	3	3	3	3
α-starch	7	7	7	7	7	7
Fish oil	3	3	3	3	3	3
Corn oil	6.4	6.4	6.4	6.4	6.4	6.4
Soy lecithin	2	2	2	2	2	2
GML	0	0.01	0.02	0.04	0.08	0.16
Ca(H_2_PO_4_)_2_-H_2_O	2.5	2.5	2.5	2.5	2.5	2.5
CaCO_3_	0.7	0.7	0.7	0.7	0.7	0.7
Alpha cellulose	1.64	1.38	1.37	1.35	1.31	1.29
Vitamins ^1^	0.75	0.75	0.75	0.75	0.75	0.75
Minerals ^2^	0.75	0.75	0.75	0.75	0.75	0.75
Y_2_O_3_	0.1	0.1	0.1	0.1	0.1	0.1
Phytase	0.05	0.05	0.05	0.05	0.05	0.05
L-carnitine	0.2	0.2	0.2	0.2	0.2	0.2
CMC	0.05	0.05	0.05	0.05	0.05	0.05
Carrageenan	0.2	0.2	0.2	0.2	0.2	0.2
DL-methionine	0.8	0.8	0.8	0.8	0.8	0.8
L-lysine	0.21	0.21	0.21	0.21	0.21	0.21
Taurine	0.5	0.5	0.5	0.5	0.5	0.5
Total	100	100	100	100	100	100
Nutrient contents ^3^						
Protein	41.50	41.77	41.71	41.90	41.89	41.83
Lipid	14. 14	14.10	14.16	14.11	13.99	14.08
Carbohydrates	27.41	27.49	27.39	27.46	27.40	27.51
Energy kJ/g ^4^	19.35	19.38	19.32	19.39	19.41	19.36
P/E ratio	1.90	1.98	2.04	2.07	2.09	2.11
P available	0.78	0.79	0.86	0.91	0.95	0.97
Total phosphorus	1.38	1.41	1.43	1.48	1.54	1.55
Calcium	0.98	0.99	0.101	0.103	0.97	0.104
Ca/P	0.71	0.73	0.74	0.79	0.83	0.74
Methionine	1.39	1.42	1.47	1.48	1.49	1.51
Lysine	2.82	2.84	2.89	2.92	2.97	2.99
Arginine	2.48	2.49	2.52	2.57	2.61	2.62
Fish oil	4.55	4.56	4.56	4.59	4.57	4.65

^1^ Vitamin premixes (mg kg^−1^ of diet): folic acid, 10; cholecalciferol, 40; menadione, 15; riboflavin 22; DL-alpha-toco-phenyl acetate, 80; niacin, 165; thiamin mononitrate, 45; vitamin B_12_, 0.04; D-Ca pantothenate, 102; ascorbic acid, 150; inositol, 450; 0.1, retinyl-acetate. ^2^ Mineral premix (g kg^−1^ of the total premix): CaCO_3_, 350; KH_2_PO_4_, 200; Cu-Cl_2_·2H_2_O, 2, NaH_2_PO_4_·H_2_O, 200; Fe-SO_4_·7H_2_O, 2; MgSO_4_·7H_2_O, 10; Sodium chloride, 12; CoCl_2_·6H_2_O, 0.1; KI, 0.1; AlCl_3_·6H_2_O, 1; sodium molybdenum oxide·2H_2_O, 0.5; and KF, 1, Na_2_SiO_3_, 0.4, MnSO_4_·H_2_O, 2, 1; Zn-SO_4_·7H_2_O, ^3^ Values for the proximate investigation of the diets are the means of the experimental/triplicate studies. ^4^ Gross energy.

**Table 2 animals-14-02963-t002:** Fatty acid composition (g/100 g) of the experimental diets fed to black sea bream.

	Control (0.00%)	GML1 (0.01%)	GML2 (0.02%)	GML3 (0.04%)	GML4 (0.08%)	GML5 (0.16%)
12:0	<LOQ	20	23	27	30	31
14:0	3.0	8.0	8.0	9.0	9.0	9.0
16:0	9.0	15	15	15	15	16
18:0	3.0	3.0	3.0	3.0	3.0	3.0
18:1n-9	36	8.0	9	17	17	17
18:1n-7	4.0	2.0	2.0	2.0	2.0	2.0
18:2n-6	15	13	11	14	15	15
18:3n-3	7.0	2.0	2.0	4.0	2.0	2.0
20:1n-9	6.0	5.0	5.0	5.0	5.0	5.0
18:4n-3	2.0	2.0	2.0	2.0	2.0	2.0
20:4n-6 ARA	0.3	0.3	0.3	0.3	0.2	0.4
22:1n-11	8.0	8.0	8.0	7.0	7.0	7.0
20:5n-3 EPA	4.0	4.0	4.0	4.0	4.0	4.0
22:5n-3 DPA	0.4	0.4	0.4	0.4	0.3	0.3
22:6n-3 DHA	5.0	5.0	5.0	4.0	4.0	4.0
Saturated FA	14	51	49	40	53	53
Sum 16:1	3.0	4.0	4.0	4.0	4.0	4.0
Sum 18:1	40	10	11	19	19	19
Sum 20:1	6.0	5.0	5.0	5.0	5.0	5.0
Sum 22:1	8.0	8.0	8.0	7.0	7.0	7.0
Sum MUFA	57	27	28	35	35	35
Sum EPA + DHA	9.0	9.0	9.0	8.0	8.0	8.0
Sum n-3	18.4	13.4	13.4	14.4	12.3	10.3
Sum n-6	15	13	11	14	15	15
Sum PUFA	33.4	26.4	24.4	28.4	27.3	25.3
n-3/n-6	1.7	1.4	1.7	1.4	0.8	0.7
n-6/n-3	0.8	0.9	0.8	0.9	1.2	1.5

Note: Fish fed control diet, 0.0%; fish fed diet GML1, 0.01%; fish fed diet GML2, 0.02%; fish fed diet GML3, 0.04%; fish fed diet GML4, 0.08%; fish fed diet GML5, 0.16%. LOQ: limit of quantification (0.0% sample); ARA: arachidonic acid; EPA: eicosapentaenoic acid; DPA: docosapentaenoic acid; DHA: docosahexaenoic acid; MUFA: mono-unsaturated fatty acids; PUFA: polyunsaturated fatty acids.

**Table 3 animals-14-02963-t003:** Effect of GML on growth performance and feed utilization of *A. schlegelii* (n = 9).

Parameters	Diets					
	Control (0.00%)	GML1 (0.01%)	GML2 (0.02%)	GML3 (0.04%)	GML4 (0.08%)	GML5 (0.16%)
IBW ^1^	1.55 ± 0.01	1.56 ± 0.02	1.56 ± 0.01	1.54 ± 0.02	1.55 ± 0.01	1.54 ± 0.01
FBW ^2^	18.00 ± 0.57 ^b^	20.15 ± 1.12 ^ab^	20.83 ± 0.45 ^a^	21.86 ± 1.12 ^a^	20.58 ± 0.06 ^a^	19.71 ± 1.83 ^ab^
WG (%) ^3^	1007.98 ± 1.51 ^b^	1008.69 ± 1.14 ^ab^	1011.03 ± 0.79 ^a^	1011.24 ± 0.58 ^a^	1010.15 ± 1.25 ^ab^	1008.55 ± 1.09 ^ab^
SGR (%/d) ^4^	4.24 ± 0.01 ^b^	4.48 ± 0.20 ^ab^	4.52 ± 0.01 ^a^	4.66 ± 0.04 ^a^	4.56 ± 0.06 ^a^	4.46 ± 0.03 ^ab^
MFI (g fish^−1^ day^−1^) ^5^	0.46 ± 0.07	0.43 ± 0.03	0.45 ± 0.15	0.32 ± 0.26	0.44 ± 0.03	0.43 ± 0.02
FCR ^6^	129.80 ± 0.47 ^a^	129.32 ± 0.87 ^ab^	128.02 ± 1.14 ^ab^	127.45 ±0.49 ^b^	129.03 ± 1.08 ^ab^	129.84 ± 0.49 ^a^
PER ^7^	1.58 ± 0.05 ^b^	1.40 ± 0.06 ^bc^	1.68 ± 0.08 ^a^	1.64 ± 0.07 ^a^	1.68 ± 0.03 ^a^	1.58 ± 0.05 ^b^
PPV (%) ^8^	30.60 ± 2.45	29.81 ± 0.50	32.01 ± 0.94	31.40 ± 0.89	29.81 ± 0.51	28.25 ± 1.65
CF (%) ^9^	2.95 ± 0.08	3.03 ± 0.14	2.92 ± 0.03	2.95 ± 0.05	2.97 ± 0.08	2.94 ± 0.08
HSI (%) ^10^	1.73 ± 0.07 ^b^	1.95 ± 0.11 ^ab^	1.87 ± 0.03 ^ab^	2.56 ± 0.40 ^a^	2.26 ± 0.39 ^ab^	1.79 ± 0.02 ^b^
IPF (%) ^11^	2.18 ± 0.22	2.19 ± 0.17	2.17 ± 0.05	2.22 ± 0.19	2.36 ± 0.26	1.86 ± 0.05
VSI (%) ^12^	7.38 ± 0.61	7.89 ± 0.22	7.63 ± 0.37	7.85 ± 0.18	8.24 ± 0.59	7.79 ± 0.20
SR (%) ^13^	99.16 ± 1.44	97.50 ± 2.50	96.66 ± 3.82	99.16 ± 1.44	98.33 ± 2.88	96.66 ± 3.82

Values are mean ± SD of three aquariums (n = 3). Values with various superscript letters in the same row are significantly different (*p* < 0.05). Abbreviations: ^1^ IBW, initial body weight; ^2^ FBW, final body weight; ^3^ WG, weight gain; ^4^ SGR, specific growth rate; ^5^ MFI, mean feed intake; ^6^ FCR, feed conversion ratio; ^7^ PER, protein efficiency ratio; ^8^ PPV, protein productive value; ^9^ CF, condition factor; ^10^ HSI, hepatosomatic index; ^11^ IPF, intraperitoneal fat ratio; ^12^ VSI, viscero-somatic index; ^13^ SR, survival rate.

**Table 4 animals-14-02963-t004:** Effects of various dietary levels of GML on the proximate composition (%) of whole body and dorsal muscle of *A. schlegelii* (n = 9).

Parameters	Diets					
	Control (0.00%)	GML1 (0.01%)	GML2 (0.02%)	GML3 (0.04%)	GML4 (0.08%)	GML5 (0.16%)
Whole body composition						
Moisture	68.24 ± 0.63 ^ab^	69.07 ± 0.45 ^a^	68.61 ± 0.74 ^ab^	66.79 ± 2.79 ^b^	67.94 ± 0.71 ^b^	68.73 ± 0.65 ^b^
Protein	48.52 ± 11.26	56.36 ± 3.88	53.92 ± 8.65	58.52 ± 1.16	53.13 ± 3.04	53.67 ± 2.37
Lipid	25.96 ± 1.18	26.11 ± 2.34	26.98 ± 1.11	28.64 ± 1.59	28.32 ± 2.19	27.05 ± 1.29
Ash	15.93 ± 0.84	16.07 ± 1.08	16.97 ± 1.76	15.35 ± 0.435	15.35 ± 0.83	15.35 ± 0.76
Dorsal muscle proximate						
Moisture	82.68 ± 11.63	79.58 ± 9.52	82.95 ± 6.93	82.57 ± 6.59	78.27 ± 7.14	79.39 ± 8.28
Protein	80.08 ± 7.48	64.42 ± 7.52	76.28 ± 11.34	76.08 ± 10.95	68.84 ± 5.03	65.67 ± 22.89
Lipid	11.61 ± 1.48	10.93 ± 0.85	10.53 ± 1.35	11.71 ± 0.22	11.18 ± 0.98	10.56 ± 1.16
Ash	9.64 ± 2.61	8.30 ± 1.75	9.58 ± 3.25	6.75 ± 0.77	7.19 ± 1.17	7.27 ± 1.26

Values are mean ± SD of three aquariums (n = 3). Values with various superscript letters in the same row are significantly different (*p* < 0.05).

**Table 5 animals-14-02963-t005:** Activities of digestive enzymes of juvenile *A. schlegelii* fed experimental diets for eight weeks (n = 9).

Parameters	Diets					
	Control (0.00%)	GML1 (0.01%)	GML2 (0.02%)	GML3 (0.04%)	GML4 (0.08%)	GML5 (0.16%)
Trypsin (U mgprot^−1^)	3032.39 ± 1374.15	2584.07 ± 528.17	3832.82 ± 2302.82	3011.35 ± 613.73	3737.15 ± 1778.44	3232.42 ± 679.53
Lipase (U gprot^−1^)	4.56 ± 0.48 ^a^	3.00 ± 0.06 ^ab^	2.78 ± 0.77 ^b^	2.67 ± 0.39 ^b^	3.01 ± 0.94 ^ab^	3.36 ± 0.78 ^ab^
Amylase (U mgprot^−1^)	3.74 ± 1.48	4.06 ± 1.47	4.42 ± 1.94	2.94 ± 1.33	3.90 ± 0.45	3.59 ± 1.33

Values are mean ± SD of three aquariums (n = 3). Values with various superscript letters in the same row are significantly different (*p* < 0.05).

**Table 6 animals-14-02963-t006:** Serum immune and biochemical parameters of juvenile *A. schlegelii* fed experimental diets for eight weeks (n = 9).

Parameters	Diets					
	Control (0.00%)	GML1 (0.01%)	GML2 (0.02%)	GML3 (0.04%)	GML4 (0.08%)	GML5 (0.16%)
TP (g L^−1^)	36.34 ± 8.72	35.37 ± 5.30	34.39 ± 12.08	37.67 ± 2.33	37.98 ± 7.69	41.44 ± 3.08
ALB (g L^−1^)	13.65 ± 3.27	11.91 ± 0.87	16.08 ± 2.96	12.74 ± 2.15	15.33 ± 1.62	13.86 ± 2.65
LZM (U mL^−1^)	66.66 ± 17.97	69.05 ± 17.97	50.00 ± 7.14	69.05 ± 14.86	52.38 ± 10.91	52.38 ± 4.12
ALT (U L^−1^)	3.34 ± 0.51	2.82 ± 0.49	2.10 ± 0.49	2.92 ± 0.92	2.29 ± 0.42	2.28 ± 0.75
AST (U L^−1^)	10.26 ± 1.56	10.28 ± 4.05	9.86 ± 3.35	7.52 ± 4.24	11.07 ± 1.93	10.33 ± 0.29
T-CHO (mmol L^−1^)	10.36 ± 3.23	9.45 ± 2.02	8.16 ± 1.78	9.62 ± 2.11	9.68 ± 2.41	10.43 ± 0.81
IgM (mg mL^−1^)	1.92 ± 0.01 ^b^	1.92 ± 0.02 ^b^	1.95 ± 0.05 ^ab^	1.99 ± 0.02 ^a^	1.923 ± 0.02 ^ab^	1.94 ± 0.03 ^ab^
C3 (μg mL^−1^)	276.09 ± 1.58 ^b^	276.33 ± 0.75 ^b^	279.05 ± 0.85 ^ab^	279.94 ± 1.23 ^a^	278.04 ± 1.66 ^ab^	276.01 ± 0.81 ^b^
C4 (μg mL^−1^)	138.95 ± 1.66 ^b^	139.24 ± 2.30 ^b^	142.72 ± 0.55 ^ab^	143.80 ± 1.15 ^a^	140.09 ± 0.64 ^ab^	139.44 ± 1.22 ^b^

Values are mean ± SD of triplicate aquarium (n = 3). Values with various superscript letters in the same row are significantly different (*p* < 0.05). Abbreviations: TP, total protein; ALB, albumin; SOD, superoxide dismutase; MDA, malondialdehyde; GSH-Px, glutathione peroxidase; CAT, catalase; T-AOC, antioxidant capacity; LZM, lysozyme; ALT, alanine transaminase; AST, aspartate aminotransferase; T-CHO, total cholesterol; IgM, immunoglobulin M; C3, complement protein C3; C4, complement protein C3.

**Table 7 animals-14-02963-t007:** Serum immune and biochemical parameters of juvenile *A. schlegelii* fed experimental diets for eight weeks (n = 9).

Parameters	Diets					
	Control (0.00%)	GML1 (0.01%)	GML2 (0.02%)	GML3 (0.04%)	GML4 (0.08%)	GML5 (0.16%)
Hindgut						
TP (g L^−1^)	3.19 ± 0.36	3.38 ± 0.14	3.47 ± 0.54	3.49 ± 0.35	3.20 ± 0.09	3.33 ± 0.16
ALB (g L^−1^)	20.32 ± 0.14	20.67 ± 0.5	20.39 ± 0.84	20.51 ± 0.19	20.42 ± 0.24	20.17 ± 0.64
LZM (U mL^−1^)	15.94 ± 6.14	16.03 ± 3.13	14.01 ± 0.75	11.78 ± 5.52	11.30 ± 2.50	16.32 ± 1.95
T-CHO (mmol L^−1^)	6.50 ± 0.08	6.51 ± 0.13	6.41 ± 0.12	6.43 ± 0.11	6.41 ± 0.06	6.64 ± 0.26
Midgut						
TP (gL^−1^)	2.50 ± 0.27	2.51 ± 0.11	2.25 ± 0.09	2.42 ± 0.04	2.42 ± 0.15	2.46 ± 0.24
ALB (g L^−1^)	18.53 ± 0.42	18.72 ± 0.15	18.79 ± 0.15	18.84 ± 0.24	18.74 ± 0.08	18.79 ± 0.20
LZM (U mL^−1^)	18.62 ± 0.79	17.42 ± 5.15	17.77 ± 4.48	19.29 ± 2.59	17.96 ± 5.03	14.79 ± 1.11
T-CHO (mmol L^−1^)	10.53 ± 1.14	10.05 ± 1.43	8.47 ± 0.82	9.49 ± 1.26	8.73 ± 2.43	8.54 ± 0.89

Values are mean ± SD of three aquariums (n = 3). Abbreviations: TP, total protein; ALB, albumin; LZM, lysozyme; T-CHO, total cholesterol.

**Table 8 animals-14-02963-t008:** Effect of different dietary levels of GML on structure of fore intestinal mucosa in juvenile black sea bream, *A. schlegelii* (n = 9).

Parameters	Diets					
	Control (0.00%)	GML1 (0.01%)	GML2 (0.02%)	GML3 (0.04%)	GML4 (0.08%)	GML5 (0.16%)
VH (μm)	435.82 ± 3.02 ^c^	464.22 ± 8.88 ^c^	505.35 ± 2.09 ^ab^	517.53 ± 3.63 ^a^	488.11 ± 15.35 ^b^	437.58 ± 9.96 ^c^
CD (μm)	95.56 ± 1.17 ^ab^	96.37 ± 1.88 ^a^	98.06 ± 3.19 ^a^	97.99 ± 3.02 ^a^	88.19 ± 2.09 ^bc^	85.27 ± 4.57 ^c^
VH/CD (μm)	7.12 ± 0.18 ^b^	7.60 ± 0.22 ^ab^	6.87 ± 0.20 ^b^	8.15 ± 0.47 ^a^	7.61 ± 0.52 ^ab^	7.28 ± 0.25 ^ab^
GC/VH	15.18 ± 0.73 ^b^	19.17 ± 0.13 ^ab^	19.05 ± 0.29 ^ab^	24.66 ± 0.68 ^a^	23.42 ± 0.43 ^a^	18.97 ± 5.16 ^ab^

Values are mean ± SD of three aquariums (n = 3). Values with various superscript letters in the same row are significantly different (*p* < 0.05). Abbreviations: VH, Villus height; CD, Crypt depth; VH/CD, Villus height/Crypt depth; GC/VH, Number of goblet cells/Villus height.

## Data Availability

The original contributions presented in the study are included in the article; further inquiries can be directed to the corresponding authors.

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
