# Peer review of "Effect of Dietary Supplementation of Glycerol Monolaurate on Growth Performance, Digestive Enzymes, Serum Immune and Antioxidant Parameters, and Intestinal Morphology in Black Sea Bream"

_animals, 2024, doi:10.3390/ani14202963_

Round 1
Reviewer 1 Report (Previous Reviewer 2)
Comments and Suggestions for Authors
The manuscript Effect of Dietary Supplementation of Glycerol Monolaurate on Growth Performance, Digestive Enzymes, Serum Immune and Antioxidant Parameters, and Intestinal Morphology in Black Sea Bream, by Ullah et al., presents a necessary topic for the aquaculture sector.
Although the authors did not attach answers to the questions asked, they did mark in a different color the adjustments made. The adjustments made generate congruence between the introduction, objective and conclusion as the central axis of the research.
In addition, the results and discussion shown are well presented and are sufficient to provide an understanding of the physiological processes involved in the addition of MLG to diets, as well as to provide guidelines for future research, as also mentioned by the authors.
The manuscript is of high quality and from my perspective, has enough to be accepted for publication.
Author Response
Reviewer 1.
Comment: The manuscript Effect of Dietary Supplementation of Glycerol Monolaurate on Growth Performance, Digestive Enzymes, Serum Immune and Antioxidant Parameters, and Intestinal Morphology in Black Sea Bream, by Ullah et al., presents a necessary topic for the aquaculture sector.
Although the authors did not attach answers to the questions asked, they did mark in a different color the adjustments made. The adjustments made generate congruence between the introduction, objective and conclusion as the central axis of the research.
In addition, the results and discussion shown are well presented and are sufficient to provide an understanding of the physiological processes involved in the addition of MLG to diets, as well as to provide guidelines for future research, as also mentioned by the authors.
The manuscript is of high quality and from my perspective, has enough to be accepted for publication.
Response: Thank you so much for your positive comments and recommendation of our manuscript for publication in Animal Journal after revision.
Reviewer 2 Report (Previous Reviewer 3)
Comments and Suggestions for Authors
Although the MS has been largely improved, I still think it is not appropriate to include the antioxidant parameters. There is no clear hypothesis that MCFA could influence the antioxidative status.
Author Response
Reviewer 2.
Comment: Although the MS has been largely improved, I still think it is not appropriate to include the antioxidant parameters. There is no clear hypothesis that MCFA could influence the antioxidative status.
Response: Thank you very much for such a valuable suggestion. The antioxidant parameters are deleted from the revised manuscript.
Comment: Does the introduction provide sufficient background and include all relevant references? Must be improved
Response: We appreciate your valuable feedback regarding the introduction. This was rewritten as per suggested by the reviewer.
This manuscript is a resubmission of an earlier submission. The following is a list of the peer review reports and author responses from that submission.
Round 1
Reviewer 1 Report
Comments and Suggestions for Authors
This manuscript described the potential role of Glycerol Monolaurate on fish growth. Digestive enzymes as well as serum immune responses and also intestinal morphology in Black Sea Bream. The experiment is well designed, and the results are easy to interpret. This gives readers an example that more interesting about the side by side of effects when using additives to improve the fish growth and health. The overall writing and logical explanation are well presented in this context. However, there are some minor errors need to be revised carefully before publication.
Line 27, why the ‘IBW’ should be included in the abstract to emphasize the result? IBW should be designed initially before the feeding trial. There is no need to show this index.
Line 31-32, to emphasize the significant result like C3, C4, IgM, P value should be included in the sentences.
Table1&Line121, Nutrient content3 and its note. Authors should show the tested diets composition rather than the designed level. As I see in this table, all the nutrients value shows the same level compared to control group. It is unbelievable that all the groups have been tested as same values. The calculated level show be emphasized in the notes if some nutrients values are not easy to test.
Figure1&Line 276, it is recommended that to rule out the background to show the intestinal histology pics more clearly.
Figure2&Line 284, the group (e) and (f) are same pics. There are definitely something wrong to present. Please check and revise it carefully.
Reviewer 2 Report
Comments and Suggestions for Authors
The manuscript Effect of Dietary Supplementation of Glycerol Monolaurate on Growth Performance, Digestive Enzymes, Serum Immune and Antioxidant Parameters, and Intestinal Morphology in Black Sea Bream, by Ullah et al., presents a necessary topic for the aquaculture sector, with the search for supplements that reinforce the immune system and serve as a substitute for antibiotics. In general, the manuscript is a good proposal, the introduction justifies the development, the methodology needs some adjustments, the results are abundant and in general are well presented and the discussion is good. However, some details have been detected and are mentioned below.
Line 92-94. The proposed objective includes evaluating GML to substitute antibiotics, but the experimental design does not support this objective, since there are no comparative diets with antibiotic levels to compare the effects to be determined. Although parameters focused on the immune system are determined, there is no comparative antibiotic element, especially not a test by stressor control challenge. The authors may aim to characterize the effect of GML on the immune system, but they cannot prove whether it is equal to or better than the antibiotic or whether it works against pathogens.
Line 107. Missing closing parentheses
Table 1. Was alpha cellulose used as a compensating filler for the inclusion of GML? That should be clear in the brief. Why does alpha cellulose decrease 0.35% while GML increases only 0.16%?
Line 174-175. Commas are doubled
Line 170-171. Develop briefly how the homogenates from which the supernatants were obtained were made.
Line 167 -181. This section is not part of section 2.4 Sample collection, perhaps another chemical analysis section.
Line 218. Makeup? Maybe proximal profile, bromatological profile, nutritional profile, etc. The word makeup is used in various times, English edition is missing.
231. It is established that lipase does not present differences; however, the table indicates differences by superscript.
The authors report positive effects from the inclusion of GML in diets, however, the formulation shows high amounts of SBM, which compromises the integrity of the digestive and immune system, known to cause various morphophysiological problems by consumption in aquatic organisms. Also, there are studies with approaches to mitigate the negative effects of soybean using dietary additives. That said, ¿will it be the case in this research that 43.5% of dietary SBM compromises the organism and the GML is mitigating the negative effects and restoring a state of homeostasis? ¿Is anything known in the species or in closely related species about negative effects due to high soybean intake? Discuss the topic
The conclusion does consider that the effect is under diets with high SBM, however, this does not agree with the introduction, where the idea of counteracting antibiotics is sold. It is necessary that the introduction, which gives the focus and the objective, agrees with the conclusion.
Comments on the Quality of English LanguageThe manuscript needs to be edited
Reviewer 3 Report
Comments and Suggestions for Authors
The MS titled “Effect of Dietary Supplementation of Glycerol Monolaurate on Growth Performance, Digestive Enzymes, Serum Immune and Antioxidant Parameters, and Intestinal Morphology in Black Sea Bream” evaluated the efficacy of medium-chain fatty acids (MCFAs) as fish health and growth promoters. Despite some interesting findings, major shortcomings existed in this study.
Firstly, the authors even don’t know how to cite a reference correctly. There were many citations including both first name initials and surnames, like (Z. Wang et al., 2022), (Y. Han et al., 2018), (J. Wang et al., 2015), (N. Dierick, J. Decuypere, & I. J. A. o. A. N. Degeyter, 2003b), (Z. Jiang et al., 2018), (T. Liu, Li, Li, & Feng, 2020), (H. Wang, Li, Ma, & Ding, 2021), (Y. Wang, Zhong, Wang, & Feng, 2021), (Y. Wang, Li, Zhang, & Feng, 2019), (H. Jiang, 2021), and (Fan Zhou et al., 2010). There were so many of them. These are not just format issues. These mistakes indicate that the authors lack basic scientific training. I never see a citation including first name initials.
Secondly, in the introduction part, the authors did not well describe how MCFAs work, i.e., the mechanisms.
If the MCFA was studied, the fatty acid composition of diets must be provided. There is C12:0 in diets naturally. We need to know how the experimental treatments affect the dietary C12:0 levels.
Table 3 was not on wet basis, but the caption said that the composition was on wet basis.
I don’t know how MCFAs influence the antioxidative capacity of animals. Why do we need to test the antioxidative capacity? If we have MDA, why do we need other relevant parameters such as SOD, CAT and …….
Figure 1 was of low quality. It shows no important information.
